# Evaluation of Platelet Alloimmunization by Filtration Enzyme-Linked Immunosorbent Assay

**DOI:** 10.3390/diagnostics13101704

**Published:** 2023-05-11

**Authors:** Tzong-Shi Chiueh, Hsin-Yao Wang, Min-Hsien Wu, Yu-Shan Hsueh, Hui-Chu Chen

**Affiliations:** 1Department of Laboratory Medicine, Linkou Chang Gung Memorial Hospital, Taoyuan City 333, Taiwan, China; 2School of Medicine, Chang Gung University, Taoyuan City 333, Taiwan, China; 3PhD Program in Biomedical Engineering, Chang Gung University, Taoyuan City 333, Taiwan, China; 4Graduate Institute of Biochemical and Biomedical Engineering, Chang Gung University, Taoyuan City 333, Taiwan, China

**Keywords:** antiplatelet antibody, fELISA, solid-phase red cell adherence test (SPRCA)

## Abstract

The current methods for detecting antiplatelet antibodies are mostly manual and labor-intensive. A convenient and rapid detection method is required for effectively detecting alloimmunization during platelet transfusion. In our study, to detect antiplatelet antibodies, positive and negative sera of random-donor antiplatelet antibodies were collected after completing a routine solid-phase red cell adherence test (SPRCA). Platelet concentrates from our random volunteer donors were also prepared using the ZZAP method and then used in a faster, significantly less labor-intensive process, a filtration enzyme-linked immunosorbent assay (fELISA), for detecting antibodies against platelet surface antigens. All fELISA chromogen intensities were processed using ImageJ software. By dividing the final chromogen intensity of each test serum with the background chromogen intensity of whole platelets, the reactivity ratios of fELISA can be used to differentiate positive SPRCA sera from negative sera. A sensitivity of 93.9% and a specificity of 93.3% were obtained for 50 μL of sera using fELISA. The area under the ROC curve reached 0.96 when comparing fELISA with the SPRCA test. We have successfully developed a rapid fELISA method for detecting antiplatelet antibodies.

## 1. Introduction

Transfusion-induced alloimmunization refers to the development of an immune response against foreign antigens present in transfused blood components, such as red blood cells, platelets, or plasma. Frequent transfusions often result in alloimmunization to donor antigens of red blood cells (RBCs), platelets, and white blood cells (WBCs), or even free antigens in plasma. In chronically transfused patient populations, such as those with sickle cell anemia or thalassemia, as many as 14% to 50% of individuals are reported to be alloimmunized [1,2,3]. Pretransfusion tests are convenient for routinely screening and identifying irregular antibodies against RBC antigens in specific before transfusion to avoid hemolytic disease of the fetus and newborn (HDFN), delayed hemolytic transfusion reaction, or a notable decrease in transfused red cell survival.

Platelet antigens, including Class I human leukocyte antigen (HLA) and human platelet antigens (HPA) on the cell membrane protein molecules of transfused platelets, also cause alloimmunization and result in platelet refractoriness [4]. Primary immunization against HLAs is caused by contaminating leukocytes in the platelet product [5]. The proportion of patients with platelet refractoriness due to alloantibodies alone has been estimated at approximately 18~25 percent [6,7]. Alloimmunization to HLAs is more common than alloimmunization to the HPA system and is believed to be the primary cause of immune-mediated platelet refractoriness [8,9]. Anti-HLA antibodies in alloimmunized patients receiving induction chemotherapy for acute leukemia frequently disappeared during treatment [10,11]. The transient nature of antibody production suggests that patients need to be regularly reassessed for their need to be supported with more expensive and difficult-to-obtain compatible platelet units. Transfusion services have an important role in assuring optimal responses to platelet transfusions by assessing patients for platelet refractoriness and offering HLA-matched or crossmatch-compatible platelets for patients who are alloimmunized.

The available tests for detecting antiplatelet antibodies include the solid-phase red cell adherence test (SPRCA), monoclonal antibody solid-phase platelet antibody test (MASPAT), monoclonal antibody-specific immobilization of platelet antigens assay (MAIPA), PAKPLUS test, PAKLx test, and flow cytometry (FC) [12,13,14,15,16,17]. FC, SPRCA, and SPRCA are whole-platelet-based tests. The other methods apply either platelet lysates or synthetic proteins as capture antigens [4,15]. The current clinical algorithm only recommends executing complex antiplatelet tests to reveal the immune refractoriness of platelet transfusion after at least two successive poor correct count increments (CCIs) [18]. Several formulae are used to assess platelet refractoriness; however, the 1-hour corrected count increment (CCI) is generally used as the objective measure of whether a patient is refractory to transfusion [19]. Using this formula, the expected increment depends on the number of platelets transfused and the patient’s body surface area. A 1-hour CCI less than 5~10 × 10^9^ /L or a percentage platelet recovery (PPR) less than 20% suggests platelet refractoriness [20]. Blood banks should then confirm the possible platelet refractoriness by detecting antiplatelet antibodies. Once anti-PLT antibodies are detected in a patient needing PLT transfusion, the blood transfusion laboratory may apply different policies. Either platelets from HLA-matched donors or laboratory cross-matched platelet products must be selected for the next platelet transfusion.

However, blood banks seldom screen and identify antiplatelet antibodies for patients receiving frequent platelet transfusions because of the high cost, labor-intensive nature, and complexity of the currently available tests. Without detecting antiplatelet antibodies and then providing the necessary HLA- or HPA-matched platelets, patients with severe thrombocytopenia are continuously exposed to risks of spontaneous hemorrhage. Alloimmunized patients with anti-HLA or anti-HPA antibodies are possibly refractory to subsequent platelet transfusions and present poor CCI after platelet transfusion [4,6,7,21,22]. Instead of calculating CCI twice after transfusion for two times, aggressive pretransfusion testing of the alloimmunization status and then initiating the selection of crossmatched platelets for the next transfusion are better methods to avoid ineffective subsequent platelet transfusions [4]. On one hand, the transfusion of crossmatch-incompatible platelets results in 70~100% poor CCIs, and the transfusion of crossmatch-compatible platelets results in adequate CCIs from about 80% to 92% of the time in selected patients lacking nonimmune causes of platelet refractoriness [23]. On the other hand, crossmatch-compatible platelets survive for 3.5~8.7 days, while incompatible platelets survive for only 0.1~2.4 days [24].

Nearly 50% of patients with human leukocyte antigen (HLA) antibodies manifest refractoriness to platelet transfusions [4]. Therefore, pretransfusion screening for antiplatelet antibodies and crossmatching for compatible platelet transfusion are always encouraged [25]. However, labor-intensive manual antiplatelet tests and high-cost flow cytometry are seldom applied for the pretransfusion crossmatching of platelets, even though they are clinically manageable tests.

Conventionally, an enzyme-linked immunosorbent assay (ELISA) is one of the better methods to detect antibodies because it is easy, rapid, and has high sensitivity. Antigens for ELISA detection of antibodies usually include plastic-fixable substances such as soluble proteins and DNA. Therefore, fixable antigens must be extracted from platelets by methods of sonication, treatment with detergents, and so on. Extraction processes are always accompanied by the destruction of platelets, and extracts contain not only cell surface antigens but also cellular ingredients such as cytoplasmic proteins. To overcome these obstacles, filtration ELISA (fELISA) provides a qualitative and specific technique for measuring antibodies against antigens on the surface of bacteria without extracting antigens from platelets. This technique is widely applicable to the assay of antibodies in various samples, including sera and fecal extracts, against various kinds of bacteria [26].

We applied minimally washed intact platelets as antigens for fELISA to detect anti-platelet antibodies. The test design used in this study could detect antiplatelet antibodies in only 45 min. 

## 2. Materials and Methods

### 2.1. Sera Collection from Patients and Platelet Preparation

A total of 19 positive and 13 negative sera of antiplatelet antibodies were collected after completing routine SPRCA in Linkou Chang Gung Memorial Hospital in Taiwan. For platelet preparation, eight units of platelet concentrate from random volunteer donors were obtained from the Blood Donation Center Hsinchu, Taiwan Blood Services Foundation. Dithiothreitol (Cyrusbioscience; cat. no. 101-3483-12-3) and Papain (Sigma-Aldrich; cat. no. P3125) were used for preparing the ZZAP solution. The platelet concentrates were separately and simultaneously prepared for testing using the ZZAP method [27,28]. ZZAP is a mixture of a sulfhydryl reagent (dithiothreitol) and a proteolytic enzyme (papain or ficin). This reagent was used to dissociate IgG and complement from red blood cells. We extended its application for preparing IgG and complement-free platelets. Briefly, platelet concentrates were isolated via two-step centrifugations (150× *g*, 10 min; 1000× *g*, 5 min). Plasma from each platelet concentrate was preserved as a self-control. Each platelet pellet was then washed with 0.2% bovine serum albumin (BSA; Ortho; cat. no. BA999A) in saline of pH 7.0, containing 0.009 M dipotassium EDTA. After resuspending the washed platelet pellet with saline, twice the volume of ZZAP was added to each resuspended platelet concentrate and incubated at 37 °C for 30 min. Then, the ZZAP-treated platelets were washed three times with saline via high-speed centrifugation (1000× *g*, 5 min). It took a total of 1.5 h to complete ZZAP treatment. Afterward, the ZZAP-treated platelets were resuspended in an adjusted final concentration of 5 × 10^5^ /μL in saline and then stored at 4 °C for 2 months at most.

### 2.2. Solid-Phase Red Cell Adherence Test (SPRCA)

According to the platelet crossmatch protocol, samples from patients suspected as refractory to platelet transfusions with at least two successive poor CCIs after platelet transfusion were routinely analyzed using blood bank samples and the SPRCA test described in a previous report [12]. SPRCA is a screening test able to detect IgG antibodies against certain HLA class I antigens and human platelet antigens by providing platelets from about 8 to 10 local blood group O donors with varying antigen structures. Both HLA and HPA types of those 8~10 local blood group O donors were selected to include the most frequent alleles of the Taiwanese population. Platelet concentrate product preparations were obtained from local blood establishments. Platelet-rich plasma (PRP) was obtained by centrifuging the platelet concentrate products at 150× *g* for 10 min. For the SPRCA method, fifty microliters of a 20 μg/mL concentration of antithrombocyte globulin (Dako-Patts, Copenhagen, Denmark) in pH 10.0 carbonate bicarbonate buffer were added to round-bottomed 2 × 8 microwell strips (NUNC, Roskilde, Denmark). The strips were incubated overnight at 4 °C before use. These could be stored for use for at least 2 months in the pH 10.0 buffer at 4 °C. Platelet monolayers were established by discarding the buffer, rinsing the wells with PBS, and adding 50 μL of PRP to the required number of wells. The strips were centrifuged at 890× *g* for 3 min, and, after discarding the supernatant, they were incubated at 37 °C for 15 min. Non-adherent platelets were washed off with PBS containing 0.2% bovine albumin (PBS-BSA). One hundred microliters of 1.9% glycine solution and 50 μL test serum were added to the platelet coated wells and incubated for 15 mins at 37 °C. The supernatant was discarded, and the wells were washed four times with PBS-BSA. Fifty microliters of a 0.4% suspension of group O Rh(D) positive red cells, sensitized with anti-D, together with 50 μL of anti-IgG (Commonwealth Serum Laboratories, Melbourne, Australia) were added. The latter was raw product diluted optimally with 1% fetal calf serum (FCS) in imidazole-buffered saline of pH 7.0. The optimum dilution of the anti-IgG was established by selecting the highest dilution in increments of 10 (i.e., 1:10, 1:20, 1:30, etc.) to produce a maximum agglutination strength of the sensitized red cells prepared as above. The strips were centrifuged at 890× *g* for 2 min and examined for either effacement (positive result) or a button of agglutinated red cells (negative result). Weak reactivity resulted in a partially haloed, formed button.

### 2.3. Filtration Enzyme-Linked Immunosorbent Assay (fELISA)

An easy and rapid enzyme-linked immunosorbent assay (ELISA) system, filtration ELISA, for the detection of antibodies against bacterial cell surface antigens was developed using a 96-well filtration plate fitted with a 0.22 μm membrane [26]. Instead of a 96-well filtration plate, however, this study used a customized 10-well filtration device fit individually with ten 25 mm diameter and 0.22 μm pore size hydrophilic polyvinylidene fluoride (PVDF) Durapore membrane filters (Millipore; cat. no. GVWP02500). A washing buffer used in the assay used the same 0.2% BSA EDTA saline. The ZZAP-treated platelet suspensions in a volume of 100 μL (total of 5 × 10^7^ platelets per well) were first incubated with 200 μL of blocking buffer supplemented with 5 μL or 50 μL of self-control or patient sera at 37 °C for 15 or 30 min. The above mixture was then applied over each filter of the filtration device’s wells. After being subjected to vacuum filtration with a negative pressure air pump, the whole platelet antigens are washed three times via vacuum filtration through a filter and resuspended in a 2 mL washing buffer each time. Then, the platelets were incubated with 500 μL of 1:5000 alkaline-phosphatase-labeled goat anti-human immunoglobulin antibodies (Sigma-Aldrich; cat. no. A1543), diluted with ddH_2_O at room temperature (RT) for 15 or 30 min. The whole platelet antigens were again washed three times with a 2 mL washing buffer via vacuum filtration through a filter. After thorough vacuum drying, 200 μL of nitro blue tetrazolium/5-bromo-4-chloro-indolyl-phosphate (NBT/BCIP, Sigma-Aldrich; cat. no. B1911) was applied onto each filter, and the filters were retained at RT for 10 min. Chromogen generation was stopped by vacuum washing with 1 mL ddH_2_O twice. Each filter was then disassembled from the vacuum device. Afterward, all filters were arrayed in order and digitally scanned for the final image.

### 2.4. Quantification of the Reaction Images

All digital images were processed using ImageJ software 1.35e (ImageJ, National Institutes of Health, Bethesda, MD, USA) to quantify the intensity of final chromogen reactivity for quantifying the dark intensity in each circular area of the filter membranes. The details of the operating procedure followed an illustrated tutorial video on YouTube (https://www.youtube.com/watch?v=JlR5v-DsTds (accessed on 11 January 2021)).

### 2.5. Statistical Analysis

ROC curve analysis was applied to measure the diagnostic accuracy of the fELISA chromogen intensity [29]. Sample size requirements, diagnostic accuracies, and the optimal cut-point value of reactivity ratios were determined using MedCalc Version 19.8.

## 3. Results

### 3.1. Reversible Background fELISA Reactivities of Original Platelets and ZZAP-Treated Platelets

After directly incubating with the secondary antibody in fELISA, the platelets of the randomly selected donors all demonstrated high chromogen backgrounds (intensity: 14,019.68) in the mock control (Figure 1). This background chromogen diminished to 56 ± 8% of initial intensity after one (a, a′) or two instances (b, b′) of ZZAP elution at 7 °C for 30 min. Then, the background chromogen intensity was regained after incubating the self sera with the ZZAP-treated platelets in the first antibody reaction of fELISA (A, A′ and B, B′). Afterward, the chromogen background (intensity: 18,940.63) of the whole platelets sample was enhanced by 35% after incubating with self sera (negative control). Chromogen intensities of A, A′, B, and B′ were 16,281.92, 13,570.56, 14,254.05, and 19,681.19, respectively. Using the negative control membrane as the denominator for calculating the reactivity ratios, the average reactivity ratio of the A, A′ and B, B′ membranes was 0.84 ± 0.14.

### 3.2. SPRCA-Negative Sera Presenting Similar Lower fELISA Reactivity Ratios

A total of eight platelet concentrates from random donors were pooled together and treated with the ZZAP method. The final platelet concentration was adjusted to 5 × 10^5^ /μL and used as a source of antigens for fELISA. Eight SPRCA-negative sera were randomly selected to individually test against 100 μL of platelet preparation at 37 °C for 15 min. All eight SPRCA-negative sera replenished the background intensity of the ZZAP-treated platelet control. However, they only presented a final chromogen intensity similar to the negative control. As a result of using the chromogen intensity of the negative control membrane as the denominator for calculating reactivity ratios, the average reactivity ratio of SPRCA-negative sera was 0.97 ± 0.17 (Figure 2).

### 3.3. Reactivity Ratios of fELISA Determining SPRCA Results Consistently

Another 19 SPRCA-positive and 5 SPRCA-negative sera were then tested at least twice for fELISA. Briefly, a single batch of ZZAP-treated platelets was applied to all experiments being performed across two months, from 11 January 2021 to 10 March 2021. Both mock and negative control (NC) sera were included in each run of 10-well fELISA. After storing ZZAP-treated platelets at 4 °C for 2 months, the same 24 test samples were tested in three separate runs: two runs on 20210309 and one run on 20210310. The density value of the mock control remained lower than that of the NC serum (Mock/NC < 1). The final positive interpretation was defined as a Sample/NC density ratio greater than 1. Twenty-two of twenty-four samples presented the same qualitative results as the SPRCA results. Only two samples were discrepant with the SPRCA results. Therefore, we concluded that the performance of fELISA remained consistent, even when using the platelets treated two months prior. The representative picture of final fELISA reactivity is shown in Figure 3. One ZZAP-treated platelet (no. 1) and a self-serum-incubated platelet (no. 6) were used as the mock and negative controls in each round of fELISA, respectively. A total of 24 sera were separately tested in three rounds of fELISA. Overall, five SPRCA-negative sera (sample nos. 1, 2, 9, 17, and 18) showed lower chromogen intensity than the negative control. The other 19 SPRCA-positive sera displayed variably higher chromogen intensities than the negative control. After calculating the reactivity ratios of each sample, the ROC curve of the fELISA reactivity ratio was analyzed to predict the SPRCA results. As for the 5 μL of test serum applied in each fELISA (Figure 4A), the sensitivity and specificity of diagnosis accuracy values were only 84.1% and 91.7%, respectively. Better sensitivity (93.9%) and specificity (93.3%) were obtained by adding 50 μL of test serum to each fELISA (Figure 4B), although the cutoff of the reactivity ratio remained similar (1.2722 for 5 μL, 1.2484 for 50 μL). Overall, the AUC of the ROC increased from 0.936 to 0.96 by adding test sera diluted 10-fold to fELISA.

## 4. Discussion

Currently, there is no whole-platelet-based ELISA available for testing antiplatelet antibodies. The endogenous peroxidase activity of platelets is likely to result in a universal strong background chromogen signal in fELISA. The protocol followed, therefore, switched to alkaline phosphatase and the NBT/BCIP system. However, the moderate background chromogen intensity of fELISA was still generally visible, even for original platelets. As the low-affinity receptor for the constant fragment of immunoglobulin G type IIa (FcγRIIa) on platelets could nonspecifically adsorb immunoglobulins in serum [30], a certain amount of immunoglobulins on the platelet surface was presumed to produce the background reactivity of the whole platelets in fELISA. This assumption was supported by the results of more than half of the background signal disappearing after ZZAP treatment and the background signal recovering after incubation with the self serum in fELISA. Variably higher background signals of some donor platelets were also observed.

After ZZAP treatment, the whole platelets became suitable for fELISA tests. A total of 13 SPRCA-negative sera presented a lower reactivity ratio compared to the self serum. Specific antiplatelet antibodies of SPRCA-positive sera could increase the nonspecific readsorption of immunoglobulins and even generate higher reactivity ratios in fELISA. However, the results of fELISA were only consistent with those of SPRCA qualitatively. Sera with a higher positive proportion of SPRCA did not indicate greater reactivity ratios of fELISA semiquantitatively. High-titer monospecific antiplatelet antibodies may result in a lower positive proportion of SPRCA but a higher reactivity ratio in fELISA. On the other hand, lower-titer multispecific antiplatelet antibodies may result in a higher positive proportion of SPRCA but a lower reactivity ratio in fELISA. The reactivity ratios of fELISA properly indicate titers of antiplatelet antibodies and may be suitable for predicting the CCIs of platelet transfusion.

The ZZAP-treated platelet preparations were stored at 4 °C. The fELISA results were still reproducible even after using ZZAP-treated platelets stored for two months. Compared to the stringent requirement of fresh platelet preparation for SPRCA, SPRCA, and FC, a longer shelf life for prepared platelets would make fELISA more practical and convenient for use in blood banks. Although the donor platelet concentrates selected for SPRCA and fELISA were not the same, a random selection of eight donors may successfully unweight and alleviate possible bias.

The results of fELISA with MAIPA, PAKPlus, or PAKLx tests were not compared because these are not whole-platelet-based assays. Generally, consistent results between whole-platelet-based and antigen-based assays are not achievable [31]. The method used in this study can detect antiplatelet antibodies in general but could not differentiate between anti-HLA and anti-human platelet antigen (HPA) antibodies.

Another limitation of our study is its small sample size. However, 5 negative and 19 positive sera of different reactivity strengths were included. For AUC 0.9, a null hypothesis value of 0.5, an α-error level of 0.05, and a β-error level of 0.05, a total of 5 negative and 18 positive samples was enough to fulfill the sample size requirement estimated using a MedCalc Version 19.8 evaluation.

Instead of the long turnaround time (4 h) for SPRCA, this study’s protocol made it easy to detect antiplatelet antibodies within a short turnaround time (45 min). Our fELISA was designed for qualitatively detecting the status of platelet alloimmunization to platelet status by using a pool of eight platelets pretreated with ZZAP. It was not designed for directly cross-matching appropriate platelets with patients. Once alloimmunized, issuing platelets from HLA-matched donors or selecting cross-matchable platelet products via flow cytometry are two methods to ensure the effectiveness of next platelet transfusion. Our fELISA test was not intended for detecting anti-platelet antibodies in transfusion-naïve ITP, HIT, or even VITT patients. Additionally, the clinical applicability of our fELISA highly depends on obtaining platelets from eight representative donors from blood establishments every two months because of the shelf-life limitation.

## 5. Conclusions

So far, complex methods for detecting antiplatelet antibodies and crossmatching platelets are only recommended after two suspicious poor CCIs following platelet transfusions without evident nonimmune-mediated platelet destruction or consumption. Only a few reference blood banks provide these complex methods. A convenient fELISA for the rapid detection of antiplatelet antibodies was developed to overcome this long-standing technical obstacle. Obtaining compatible platelet units is now possible through crossmatching with fELISA instead of simply relying on the issue of HLA-matched platelets.

## 6. Patents

United States Patent, Patent No.: US 11,231,425 B2.

Republic of China Patent, Patent No.: I672502.

## Figures and Tables

**Figure 1 diagnostics-13-01704-f001:**
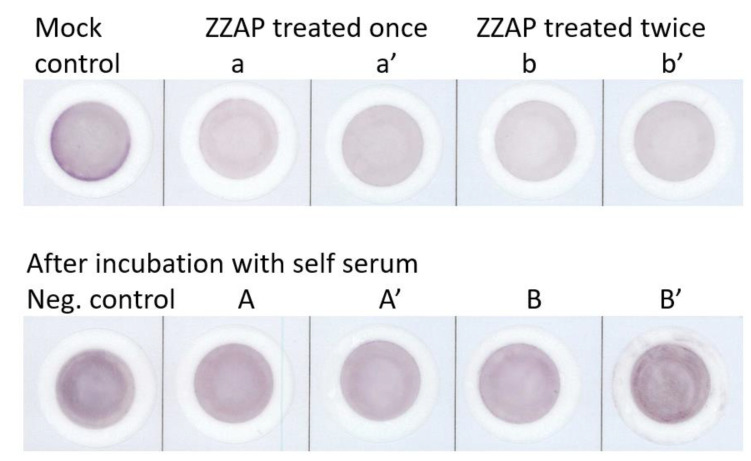
Reversible background filtration enzyme-linked immunosorbent assay (fELISA) reactivity of original platelets and ZZAP-treated platelets. Background chromogenic reactivity of original platelets (mock control) reduced after being treated with ZZAP once (a, a′) and twice (b, b′) in duplicate fELISA tests without adding serum of the first antibody. Mock control platelets reacting with self serum and processing throughout the entirety of fELISA were used as the negative control. After preincubation with self serum, ZZAP-treated platelets (A, A′ and B, B′) regained almost the same chromogen backgrounds as the mock and negative controls. All digital images were processed with according to intensity of darkness, quantified using ImageJ software.

**Figure 2 diagnostics-13-01704-f002:**
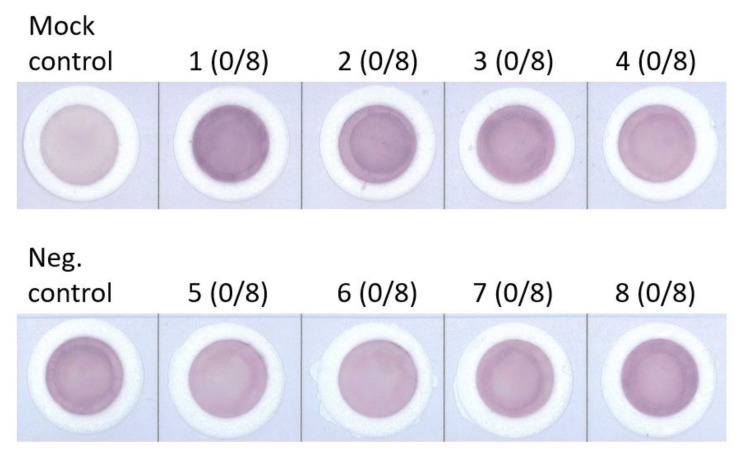
Similar lower filtration enzyme-linked immunosorbent assay (fELISA) reactivity ratios from solid-phase red cell adherence test (SPRCA)-negative sera. Both self serum (negative control) and 8 SPRCA (0/8)-negative sera (nos. 1–8) were individually tested for their reactivity with the ZZAP-treated pool platelets. All digital images were processed according to intensity of darkness, quantified using ImageJ software.

**Figure 3 diagnostics-13-01704-f003:**
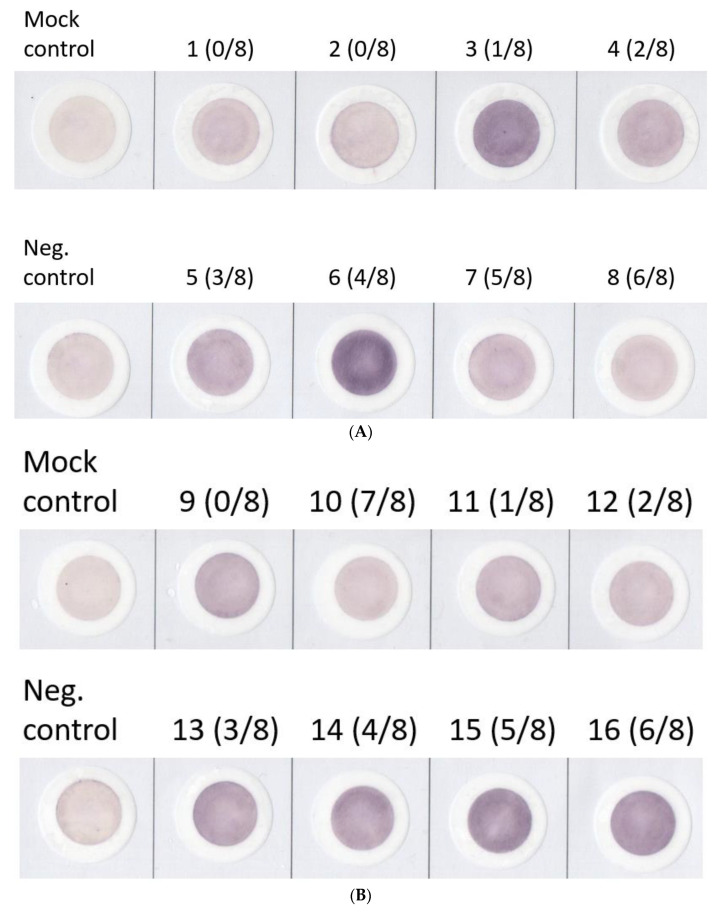
Filtration enzyme-linked immunosorbent assay (fELISA) reactivities’ congruity with the solid-phase red cell adherence (SPRCA) test results. Pooled platelets of eight different platelet concentrates were stripped via ZZAP treatment and then applied as antigens of fELISA. From a total of 24 SPRCA samples, 8 samples were tested each time ((**A**) first test; (**B**) second test; (**C**) third test) for their fELISA reactivity. Numbers in the brackets indicate the SPRCA-positive ratio of samples to eight random platelets. All digital images were processed according to intensity of darkness, quantified using ImageJ software. The negative control membrane’s reactivity was used as the denominator for calculating the reactivity ratio of each test serum.

**Figure 4 diagnostics-13-01704-f004:**
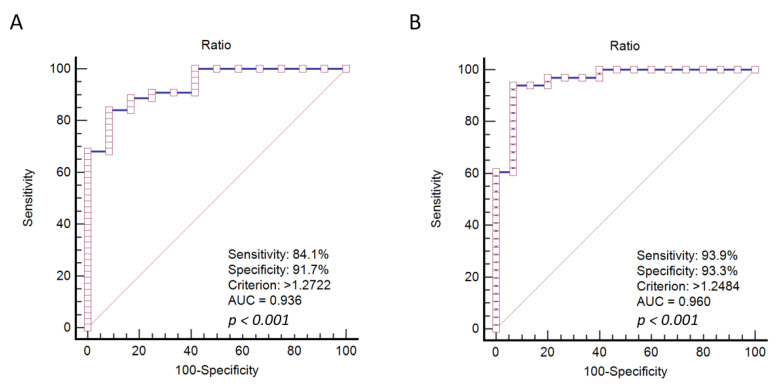
Predictions of the solid-phase red cell adherence test (SPRCA) results using the receiver operating characteristic (ROC) curve of the ratio of filtration enzyme-linked immunosorbent assay (fELISA) reactivity. The area under the ROC curve (AUC) of the ROC and statistics are indicated; *p* ≤ 0.001 is considered statistically significant. We applied (**A**) 5 uL serum and (**B**) 50 uL serum to fELISA in these instances.

## Data Availability

Data is unavailable due to confidentiality of patent.

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
