# Peer review of "Evaluation of Platelet Alloimmunization by Filtration Enzyme-Linked Immunosorbent Assay"

_diagnostics, 2023, doi:10.3390/diagnostics13101704_

Round 1

Reviewer 1 Report

The manuscript finds to be very interesting. I had gone through the whole manuscript. The results were found to be Interesting and discussion was presented well.  I had only a single query, Can authors apply this technique in which human disease as antiplatelet antibodies?

Try to improve the resolution of the figures

Please find my comments

- The main question addressed by the research is to detect antiplatelet antibodies, positive and 14 negative sera of random-donor antiplatelet antibodies were collected after completing the routine 15 solid-phase red-cell adherence test.

- This is a new technique (filtration enzyme linked immunosorbent assay) which is used to detect antibodies towards platelet surface antigens. This technique may fill the gap in this field.

- This is a new technique, time consuming and advanced technique.

- This manuscript I found to be extremely perfect and additionally authors have patent for this work

- I am quite fine with the conclusion.

- The references are appropriate.

- Both tables and figures seem to be fine

Author Response

Response to Reviewer 1 Comments

Point 1: Can authors apply this technique in which human disease as antiplatelet antibodies?

Response 1: No, this technique is only suitable for estimating the probability of encountering unmatched platelet transfusion, especially for patients having been alloimmunized with multiple platelet transfusions. And it’s not able to differentiate specific anti-platelet antibodies. So, we did not collect and test sera of ITP, HIT, or even VITT patients with our fELISA. Furthmore, these diseases were characterized more like autoimmune in pathogenesis. Obtaining enough platelets from these thrombocytopenia patients before being transfused was difficult. So, we did not test and cannot confirm whether the autoantigens were preserved after ZZAP treatment.

Appreciate for the other 7 positive comments.

Reviewer 2 Report

The manuscript by Chiueh et al presents a method for fast detection of anti-platelet antibodies in serum. The methodology is of interest and they highlight an application in transfusion medicine (as to monitor or predict potential refractoriness due to alloimmunization).

I have minor comments to the authors:

- It would be nice to briefly explain conditions of the ZZAP procedure or to refer to another publication where it is described.

- Figure 3 has pictures that have been stretched, please keep proportions.

- It would be nice to show the quantification values on a table, of the quantification with the different substrates applied (it is mentioned in the discussion, but the results are not presented really, in a clear way) and very important, show values with the tests performed with recently treated ZZAP platelets, and after two months storage. This is a very important and critical point, as the authors claim the test is fast and immediate (45 min) but it requires the preparation of an essential reagent (ZZAP platelets), that takes time and needs a biological source material, and the possibility to store it, facilitates the screening later.

- It would be pertinent to mention other applications, like detection of anti-platelet antibodies in Immune Thrombocytopenia, for example. They have tested sera from patients positive for anti HPA and anti HLA antibodies, but the method could be applied to detect antibodies to other platelet antigens. Please discuss in the discussion.

Author Response

Response to Reviewer 2 Comments

Point 1: It would be nice to briefly explain conditions of the ZZAP procedure or to refer to another publication where it is described.

Response 1: The ZZAP procedure was originally used to treat DAT positive RBCs for the subsequent autoadsorption test.  It’s reference was coded on Line 113 #27. Instead of treating RBCs, similar procedure for preparing ZZAP-treated platelets as reagent of fELISA was described on Line 117~121, “After resuspending the washed platelet precipitate with saline, twice the volume of ZZAP was added to each resuspended platelet concentrate and incubated at 37°C for 30 min. Then, the ZZAP-treated platelets were washed with saline by low-speed centrifugation. Afterward, they were resuspended with the platelet concentration quantified to 5×105/ml in saline and then stored at 4°C for 3 months at most.

Point 2: Figure 3 has pictures that have been stretched, please keep proportions.

Response 2: Three original pictures were used to avoid of stretching them instead of fitting them together in one picture.

Point 3: It would be nice to show the quantification values on a table, of the quantification with the different substrates applied (it is mentioned in the discussion, but the results are not presented really, in a clear way) and very important, show values with the tests performed with recently treated ZZAP platelets, and after two months storage. This is a very important and critical point, as the authors claim the test is fast and immediate (45 min) but it requires the preparation of an essential reagent (ZZAP platelets), that takes time and needs a biological source material, and the possibility to store it, facilitates the screening later.

Response 3: A single batch of ZZAP-treated platelets was applied to all experiments being performed in two months, from 2021/01/11 to 2021/03/10. Both mock and negative control (NC) serum were included in each run of 10-well fELISA. After storing ZZAP-treated platelets at 4℃ for 2 months, 24 test samples were tested in three separate runs, two runs on 20210309 and one run on 20210310. The following table presents raw dentity values of mock, NC, and 24 samples. Density value of mock control remain lower than NC seum (Mock/NC < 1). Final positive interpretation was defined as Sample/NC density ratio greater than 1. Twenty-two of 24 samples presented the same qualitative results with the SPRCA results. Only two samples were discrepant from the SPRCA results (bald and underlined). Therefore, we concluded the performance of fELISA remained consistent even using the platelets treated two months ago.

Excuse us for the confidential consideration. We decided not to disclose all details of densitometry and calculation in the manuscript.

DateRun

Well

Density

SPRCA

Sample/NC Density Ratio

Interpretation

20210309A

1

8976.2

Mock

0.5

20210309A

2

17749.2

0/8

1.0

N

20210309A

3

18401.6

0/8

1.0

N

20210309A

4

33857.1

1/8

1.9

P

20210309A

5

26646.8

2/8

1.5

P

20210309A

6

17732.3

NC

1.0

20210309A

7

33515.7

3/8

1.9

P

20210309A

8

31394.7

4/8

1.8

P

20210309A

9

27757.7

5/8

1.6

P

20210309A

10

23231.9

6/8

1.3

P

20210309B

1

10667.1

Mock

0.7

20210309B

2

20788.0

7/8

1.4

P

20210309B

3

17467.2

0/8

1.2

(P)

20210309B

4

27809.9

1/8

1.9

P

20210309B

5

22712.2

2/8

1.6

P

20210309B

6

14597.9

NC

1.0

20210309B

7

21104.1

3/8

1.4

P

20210309B

8

25550.1

4/8

1.8

P

20210309B

9

26628.8

5/8

1.8

P

20210309B

10

23072.8

6/8

1.6

P

20210310A

1

7200.0

Mock

0.5

20210310A

2

15212.8

0/8

1.0

N

20210310A

3

12517.8

0/9

0.8

N

20210310A

4

15937.7

1/8

1.0

(N)

20210310A

5

29049.3

2/8

1.9

P

20210310A

6

15487.9

NC

1.0

20210310A

7

22210.7

3/8

1.4

P

20210310A

8

27902.2

3/8

1.8

P

20210310A

9

20434.2

5/8

1.3

P

20210310A

10

17712.4

7/8

1.1

P

Point 4: It would be pertinent to mention other applications, like detection of anti-platelet antibodies in Immune Thrombocytopenia, for example. They have tested sera from patients positive for anti HPA and anti HLA antibodies, but the method could be applied to detect antibodies to other platelet antigens. Please discuss in the discussion.

Response 4: Thank you for this comment. We have addded the following descriptions on Line 326-331 : “Our fELISA test may be useful to detect anti-platelet antibodies for ITP, HIT, and even VITT as well. However, these diseases were characterized more like autoimmune in pathogenesis. Obtaining enough platelets from these thrombocytopenia patients before being transfused was difficult. So we did not test and cannot confirm whether the autoantigens were preserved after ZZAP treatment.”

Reviewer 3 Report

2282813_MDPI Diagnostics_ Chiueh

The paper by T.S. Chiueh and Coworkers describes a feasibility study of an assay aimed at the screening of anti-platelet antibodies by a filtration ELISA (fELISA) technique.

GENERAL COMMENTS

The paper describes an assay on whole platelets that is designed and can be used for cross-matching purposes only: a different thing from an assay aimed at the detection of anti-platelet antibodies, as the title indicates. The title should be reworded, to better describe the very aim of this fELISA assay, because it is unable to discriminate or characterize anti-HLA and anti-HPA antibodies.

The described fELISA is an assay using 'stripped' whole platelet preparations, kept stored for up to three months, that can be used on demand also for the SPRCA assay as the solid phase substrate for a donor-specific platelet cross-match. The feasibility of this approach seems reasonable, since the turn-around-time of the final fELISA assay is about 45 minutes for a single test. A more extensive discussion on the features and interexchangeability of the three whole platelet antibody assays developed so far (i.e. fELISA, SPRCA and Flow Cytometry) is warranted here.

To envisage a real world application of the fELISA assay, however, one may assume that a vast array of platelet targets should be available once an immunized patients has to be transfused. The number of whole platelet preparations that could be tested to tackle the transfusion requirement of an immunized patient should be estimated, and this may involve both the size of the blood transfusion facility (not all centers may be able to afford this procedure) and the additional time required to test multiple donor-recipient combinations in parallel. All these important issues were not discussed in the present paper.

The Authors should also address the management of the therapeutic platelet units (also specifying if single-donor concentrates or platelet pools can be used on the basis of the fELISA assay results), the organisation of the platelet donor involvement and all the necessary downstream details. In other words, the fELISA assay is not a stand-alone item, but should be set in a more general picture of a transfusion laboratory organisation.

The language is imprecise, with several unclear sentences and many inappropriate terms, needing a thorough review by a mother tongue expert.

SPECIFIC COMMENTS

Title: Something like "Rapid Platelet Cross-Matching by Filtration Enzyme-Linked Immunosorbent Assay" seems more appropriate.

Abstract, first line and main text: Commercial methods to detect anti-PLT antibodies cannot be honestly defined as 'too complicated' or 'complex', since most of them can be accomplished by full laboratory automation. Please reword where appropriate.

Abstract line 21 and main text at lines 186, 193, 200, 263 etc: The term 'authentic' to define whole platelets is inappropriate. Please use the 'whole' or 'native' adjectives when discussing the assays using unmodified PLT suspensions versus the tests using other PLT-derived products.

Main text: It is necessary to keep clear the PLT Cross-Matching aim of the described fELISA assay throughout the main text, to avoid confusion with other assays aimed at the detection and characterisation of anti-PLT antibodies.

Introduction, lines 32-33: The sentence "...given how current pretransfusion tests are mostly RBC ag-32 glutination-based methods." is obscure. Please reword.

Introduction, lines 69-71: Once anti-PLT antibodies are detected in a patient needing PLT transfusion, the blood transfusion laboratory may apply different policies. Please add a few lines summarizing the laboratory and clinical strategies that may be applied once a positive anti-PLT antibody screening test is recorded. 

Introduction, lines 76-78: The sentence "...could only  determine unmatched platelets after completing transfusion without pretransfusion screening and crossmatching" is really unclear. Since ref.#4 is cited here, it seems good to remind that a rapid PLT disappearance after transfusion does not necessarily mean that this is due by an immune-mediated mechanism.

Introduction, lines 86-87: The sentence  "...the complexity of current antiplatelet tests does not make them suitable for pretransfusion crossmatching platelets because of long turnaround times" is questionable, since both SPRCA and FC pre-transfusion testings are known and described as clinically manageable (see for instance: Salama OS et al. Blood Transfus 2014; 12: 187-94  and  Sayed D et al. J Clin Apheresis 2011; 26(1): 23-28). Please reword.

Material and Methods, line 108: "For preparing platelet antigens..." please reword, since whole PLTs and not antigens are prepared as reaction targets.

Material and Methods, line 111. Please briefly describe what are the function and the advantages of the ZZAP solution treatment of PLT suspensions.

Material and Methods, line 114. "...supernatant sera..." please correct as plasma.

Material and Methods, line 115. Platelet pellet, not precipitate, which is a different thing.

Material and Methods, line 121. Please conclude this chapter indicating how long does it take the preparation of an array of ZZAPped PLT suspensions, how do you have validated the 3-month shelf life and how do you check if the target ZZAPped PLT suspensions are still usable or altered during the storage at 4°C, since no fixatives are used. The same details should be added when describing the 2-month preservation of SPRCA strips, at line 136.

Material and Methods, 2.4. Quantification of the reaction images.  The ImageJ software seems designed for the quantification of gel bands only. Please describe how this software can be used on the fELISA discs, how the ratio with the blank reaction sample is calculated and what are the limits of variance of the background, which seems a major variable of this assay. 

Material and Methods, line 95. The fELISA assay is defined as a quantitative assay. Due to its delicate design, the multiple laboratory steps, the high test background and the needed calculations, it seems more practical and clinically relevant to use the fELISA assay as a qualitative methods, by simply setting a reliable cutoff level and defining the 'positive reaction' territory without any further attempt to quantify the binding reaction.

Figure 4. The legend reports inverted panel A and B. The results in the panel A refer to the usage of 5 microliter sera samples, and vice-versa.

Discussion. Please clarify the exact positioning of the fELISA assay as a PLT cross-matching test. A discussion on the other whole PLT assay (Flow Cytometry) can be also of interest here.

Main text. Many terms are used incorrectly and should be replaced by a  more appropiate wording. For example: authentic platelets, line 30 manifest, line 41 possibly, line 99 privilege, imprecise definition of 5 and 50 microliters in various points.

Author Response

Response to Reviewer 3 Comments

GENERAL COMMENTS

The paper describes an assay on whole platelets that is designed and can be used for cross-matching purposes only: a different thing from an assay aimed at the detection of anti-platelet antibodies, as the title indicates. The title should be reworded, to better describe the very aim of this fELISA assay, because it is unable to discriminate or characterize anti-HLA and anti-HPA antibodies.

SPRCA assay can not differentiate anti-HPA and anti-HLA Abs. They are belonging to tests for detecting anti-platelet Abs.

The described fELISA is an assay using 'stripped' whole platelet preparations, kept stored for up to three months, that can be used on demand also for the SPRCA assay as the solid phase substrate for a donor-specific platelet cross-match. The feasibility of this approach seems reasonable, since the turn-around-time of the final fELISA assay is about 45 minutes for a single test. A more extensive discussion on the features and inter-exchangeability of the three whole platelet antibody assays developed so far (i.e. fELISA, SPRCA and Flow Cytometry) is warranted here.

To envisage a real world application of the fELISA assay, however, one may assume that a vast array of platelet targets should be available once an immunized patients has to be transfused. The number of whole platelet preparations that could be tested to tackle the transfusion requirement of an immunized patient should be estimated, and this may involve both the size of the blood transfusion facility (not all centers may be able to afford this procedure) and the additional time required to test multiple donor-recipient combinations in parallel. All these important issues were not discussed in the present paper.

The Authors should also address the management of the therapeutic platelet units (also specifying if single-donor concentrates or platelet pools can be used on the basis of the fELISA assay results), the organisation of the platelet donor involvement and all the necessary downstream details. In other words, the fELISA assay is not a stand-alone item, but should be set in a more general picture of a transfusion laboratory organisation.

The language is imprecise, with several unclear sentences and many inappropriate terms, needing a thorough review by a mother tongue expert.

SPECIFIC COMMENTS

Point 1: Title: Something like "Rapid Platelet Cross-Matching by Filtration Enzyme-Linked Immunosorbent Assay" seems more appropriate.

Response 1: Thanks for your precise comment. The title was changed to “Evaluate platelet alloimmunization by filtration enzyme-linked immunosorbent assay”. 

Point 2: Abstract, first line and main text: Commercial methods to detect anti-PLT antibodies cannot be honestly defined as 'too complicated' or 'complex', since most of them can be accomplished by full laboratory automation. Please reword where appropriate.

Response 2: Thank you for suggesting reword. Abstract, first line was modified as “Commercial methods to detect anti-PLT antibodies are mostly manual and labor intensive.” There are no full automation system for detecting anti-PLT antibodies so far. Protocols of current available methods are more complex than our fELISA.

Point 3: Abstract line 21 and main text at lines 186, 193, 200, 263 etc: The term 'authentic' to define whole platelets is inappropriate. Please use the 'whole' or 'native' adjectives when discussing the assays using unmodified PLT suspensions versus the tests using other PLT-derived products.

Response 3: Thank you for the comment. The term ‘authentic’ was changed to ‘whole’ or ‘original’ throughout the whole manuscript.

Point 4: Main text: It is necessary to keep clear the PLT Cross-Matching aim of the described fELISA assay throughout the main text, to avoid confusion with other assays aimed at the detection and characterisation of anti-PLT antibodies.

Response 4: Thank you for the comment. However, our aim of fELISA is to evaluate the alloimmunization status of patients after experiencing multiple platelet transfusions, not to matching appropriate platelets for patients. We modified the title as your first recommendation. The aim of this manuscript should become clear and helpful for avoiding confusion. 

Point 5: Introduction, lines 32-33: The sentence "...given how current pretransfusion tests are mostly RBC ag-32 glutination-based methods." is obscure. Please reword.

Response 5: Thank you for the comment. We deleted the redundant sentence.

Point 6: Introduction, lines 69-71: Once anti-PLT antibodies are detected in a patient needing PLT transfusion, the blood transfusion laboratory may apply different policies. Please add a few lines summarizing the laboratory and clinical strategies that may be applied once a positive anti-PLT antibody screening test is recorded. 

Response 6: Thank you for the comment. We added the following sentences “Once anti-PLT antibodies are detected in a patient needing PLT transfusion, the blood transfusion laboratory may apply different policies. Either platelet from HLA-matched donors or laboratory cross-matched platelet products must be selected for the next platelet transfusion.” on Line 70-72.

Point 7: Introduction, lines 76-78: The sentence "...could only determine unmatched platelets after completing transfusion without pretransfusion screening and crossmatching" is really unclear. Since ref.#4 is cited here, it seems good to remind that a rapid PLT disappearance after transfusion does not necessarily mean that this is due by an immune-mediated mechanism.

Response 7: Thank you for the comment. We modified the description to the following “Calculating the posttransfusion absolute count or CCI of platelets were both post hoc analysis for realizing unmatched platelet transfusions, because pretransfusion screening and crossmatching were not routinely performed for platelet transfusions” on Line 81-84.

Point 8: Introduction, lines 86-87: The sentence  "...the complexity of current antiplatelet tests does not make them suitable for pretransfusion crossmatching platelets because of long turnaround times" is questionable, since both SPRCA and FC pre-transfusion testings are known and described as clinically manageable (see for instance: Salama OS et al. Blood Transfus 2014; 12: 187-94  and  Sayed D et al. J Clin Apheresis 2011; 26(1): 23-28). Please reword.

Response 8: Thank you for the comment. We modified the sentence as the following “However, the labor intensive manual antiplatelet tests and high cost flowcytometry were seldom applied for pretransfusion crossmatching platelets, although they were clinical manageable tests.” On Line 93-95.

Point 9: Material and Methods, line 108: "For preparing platelet antigens..." please reword, since whole PLTs and not antigens are prepared as reaction targets.

Response 9: Thank you for the comment. We rewrote as your suggestion, “For platelet preparation,….” On Line 117.

Point 10: Material and Methods, line 111. Please briefly describe what are the function and the advantages of the ZZAP solution treatment of PLT suspensions.

Response 10: Thank you for the comment. We added following sentences “ZZAP is a mixture of a sulfhydryl reagent (dithiothreitol) and a proteolytic enzyme (papain or ficin). This reagent was used to dissociate IgG and complement from red blood cells. We extended its application for preparing IgG and complement free platelets.” on Line 122-125.

Point 11: Material and Methods, line 114. "...supernatant sera..." please correct as plasma.

Response 11: Thank you for the comment. Two “sera” were replaced with “plasma” on Line 126.

Point 12: Material and Methods, line 115. Platelet pellet, not precipitate, which is a different thing.

Response 12: Thank you for the comment. Two “precipitate” were replaced with “pellet” on Line 127 and 129.

Point 13: Material and Methods, line 121. Please conclude this chapter indicating how long does it take the preparation of an array of ZZAPped PLT suspensions, how do you have validated the 3-month shelf life and how do you check if the target ZZAPped PLT suspensions are still usable or altered during the storage at 4°C, since no fixatives are used. The same details should be added when describing the 2-month preservation of SPRCA strips, at line 136.

Response 13: Thank you for the comment. More detail for platelet preparation was provided on Line 131-135. “Then, the ZZAP-treated platelets were washed three times with saline by high-speed centrifugation (1000 g, 5 min). It took total 1.5 hours to complete ZZAP treatment. Afterward, ZZAP treated platelets were resuspended and adjusted final concentration to 5×105/ml in saline and then stored at 4°C for 2 months at most.

And to address your comment about the shelf-life of ZZAP-treated platelets, we provided the following detail laboratory note and raw data.

A single batch of ZZAP-treated platelets was applied to all experiments being performed in two months, from 2021/01/11 to 2021/03/10. Both mock and negative control (NC) serum were included in each run of 10-well fELISA. After storing ZZAP-treated platelets at 4℃ for 2 months, 24 test samples were tested in three separate runs, two runs on 20210309 and one run on 20210310. The following table presents raw dentity values of mock, NC, and 24 samples. Density value of mock control remain lower than NC seum (Mock/NC < 1). Final positive interpretation was defined as Sample/NC density ratio greater than 1. Twenty-two of 24 samples presented the same qualitative results with the SPRCA results. Only two samples were discrepant from the SPRCA results (bald and underlined). Therefore, we concluded the performance of fELISA remained consistent even using the platelets treated two months ago.

Excuse us for the confidential consideration, we decided not to disclose all details of densitometric data and calculation in the manuscript.

DateRun

Well

Density

SPRCA

Sample/NC Density Ratio

Interpretation

20210309A

1

8976.2

Mock

0.5

20210309A

2

17749.2

0/8

1.0

N

20210309A

3

18401.6

0/8

1.0

N

20210309A

4

33857.1

1/8

1.9

P

20210309A

5

26646.8

2/8

1.5

P

20210309A

6

17732.3

NC

1.0

20210309A

7

33515.7

3/8

1.9

P

20210309A

8

31394.7

4/8

1.8

P

20210309A

9

27757.7

5/8

1.6

P

20210309A

10

23231.9

6/8

1.3

P

20210309B

1

10667.1

Mock

0.7

20210309B

2

20788.0

7/8

1.4

P

20210309B

3

17467.2

0/8

1.2

(P)

20210309B

4

27809.9

1/8

1.9

P

20210309B

5

22712.2

2/8

1.6

P

20210309B

6

14597.9

NC

1.0

20210309B

7

21104.1

3/8

1.4

P

20210309B

8

25550.1

4/8

1.8

P

20210309B

9

26628.8

5/8

1.8

P

20210309B

10

23072.8

6/8

1.6

P

20210310A

1

7200.0

Mock

0.5

20210310A

2

15212.8

0/8

1.0

N

20210310A

3

12517.8

0/9

0.8

N

20210310A

4

15937.7

1/8

1.0

(N)

20210310A

5

29049.3

2/8

1.9

P

20210310A

6

15487.9

NC

1.0

20210310A

7

22210.7

3/8

1.4

P

20210310A

8

27902.2

3/8

1.8

P

20210310A

9

20434.2

5/8

1.3

P

20210310A

10

17712.4

7/8

1.1

P

Point 14: Material and Methods, 2.4. Quantification of the reaction images.  The ImageJ software seems designed for the quantification of gel bands only. Please describe how this software can be used on the fELISA discs, how the ratio with the blank reaction sample is calculated and what are the limits of variance of the background, which seems a major variable of this assay. 

Response 14: Thank you for the comment. The following picture and two densitometric scan graphs illustrated the original results performed on 2021/3/10. We used one rectangle box to include 5 fELISA discs at one time. Densitometric scan graph of each fELISA disc roughly presented with a bell shape analog curve, but gel immages regularly presented with a digital curve. However, the area under each curve still faithfully represented the total chromogenic density of each individual disc (see the above table). We did realize the back grounds of mock and NC varied between test runs. After several ways to adjust and compare, ratio of test sample to NC chromogenic density had the best qualitative power to diffrentiate positive from negative results.

Point 15: Material and Methods, line 95. The fELISA assay is defined as a quantitative assay. Due to its delicate design, the multiple laboratory steps, the high test background and the needed calculations, it seems more practical and clinically relevant to use the fELISA assay as a qualitative methods, by simply setting a reliable cutoff level and defining the 'positive reaction' territory without any further attempt to quantify the binding reaction.

Response 15: Agree with your comment. We rewrote the fELISA is a qualitative test on Line 104.

Point 16: Figure 4. The legend reports inverted panel A and B. The results in the panel A refer to the usage of 5 microliter sera samples, and vice-versa.

Response 16: Thank you for the comment. The typo was fixed.

Point 17: Discussion. Please clarify the exact positioning of the fELISA assay as a PLT cross-matching test. A discussion on the other whole PLT assay (Flow Cytometry) can be also of interest here.

Response 17: Thank you for the comment. We deleted the following description about future study on Line 321-324, “For future studies, patients’ CCIs can be followed up after transfusing the pre-crossmatched platelets by fELISA to validate their clinical efficacy.” And replaced with the following paragraph, “Our fELISA was designed for qualitatively detecting the status of alloimmunization to platelets by using a pool of 8 platelets pretreated with ZZAP. It was not for directly cross-matching appropriate platelets for patients. Once alloimmunized, issuing platelets from HLA-matched donors or selecting cross-matchable platelet products by flow cytometry were two ways to ensure the effectiveness of next platelet transfusion.” on Line 322-333. We also modified the last paragraph of introduction, Line 110, the application in crossmatching platelets for transfusion was deleted.

Point 18: Main text. Many terms are used incorrectly and should be replaced by a  more appropiate wording. For example: authentic platelets, line 30 manifest, line 41 possibly, line 99 privilege, imprecise definition of 5 and 50 microliters in various points.

Response 18: Thank you for the comment. Authentic platelets were replaced by “whole” platelets through whole manuscript. Line 30 manifest was replaced by “result in”. Line 41 possibly was deleted. Line 99 “ took the privilege of” was changed to “applied”. 

Round 2

Reviewer 3 Report

The Authors have made progress in improving their original manuscript, having fixed a part of the items to be corrected. Some recommendations concerning critical issues of the original submission have been however ignored.

A number of points still need to be  better specified and improved.

Briefly:

1) The language has not been reviewed by a mother-tongue expert, as recommended, and the paper still contains many syntax errors, improper terms and obscure sentences. The frequent usage of past tense ('were' throughout the manuscript) has to be corrected into the present in most instances.

2) Title: Should be corrected as "Evaluation of platelet alloimmunization..."

3) The meaning of the entire sentence from "Calculating (line 81)...to transfusions (line 84)", despite a rewording is still very obscure.

4) The sentence from "More alert (line 111)...to blood banks (line 112)", is impropre and should be omitted.

5) The way to calculate the quantification of the reaction grade by ratioing and the confidence limits of readings in negative cases, previously recommended, are still lacking.

6) The validation of ZZAPped PLT preparation shelf life and the method to detect the integrity of the stored PLT samples, as previously recommended, are still missing.

7) The sentence from "The pathophysiological (line 288)...to future study (line 290)" must be omitted.

8) In this v2 the described assay has been more clearly set as a screening method using a pool of 8 platelet concentrates. The details of the necessary organisation to collect and renew with time PLT concentrates from HLA-typed donors should be added in the discussion.

9) The entire sentence from "Our fELISA (line 328)...to ZZAP treatment (line 333)" should be omitted. The use of an assay designed to detect allo-immunisataion is not recommended nor validated in the autoimmune setting. This statement is improper, also because HIT and VITT are studied by validated specific assays (i.e. PF4 and Functional flow assays) and alloantibodies are absent in transfusion-naive ITP patients.

Author Response

The Authors have made progress in improving their original manuscript, having fixed a part of the items to be corrected. Some recommendations concerning critical issues of the original submission have been however ignored.

A number of points still need to be better specified and improved.

Briefly:

  • The language has not been reviewed by a mother-tongue expert, as recommended, and the paper still contains many syntax errors, improper terms and obscure sentences. The frequent usage of past tense ('were' throughout the manuscript) has to be corrected into the present in most instances.

Response: Thanks for your comment. We did not ignore your recommendation. Our primary manuscript has been edited professionally. However, we need to find another editing service for better improvement. To address all modified places precisely, we were not able to reedit the manuscript in advance. We will send for the second professional editing after completing responses for all your scientific recommendations. Reediting might take more than the 10-day closing deadline. Please consider and agree with our appeal for reediting after reviewing.

  • Title: Should be corrected as "Evaluation of platelet alloimmunization..."

Response: Thanks for your correction. The title was modified as your recommendation.

  • The meaning of the entire sentence from "Calculating (line 81)...to transfusions (line 84)", despite a rewording is still very obscure.

Response: Thanks for your comment. The whole sentence was deleted and change to the following: “Instead of calculating CCI after transfusion for two times, aggressively pretransfusion testing the alloimmunization status and then initiating selection of crossmatched platelets for the next transfusion are better ways to avoid of ineffective subsequent platelet transfusions [4].”on Line 81-84. As we mentioned “The current clinical algorithm only recommends executing complex antiplatelet tests to reveal the immune refractoriness of platelet transfusion after at least two poor successive correct count increments (CCIs) [18].”on Line 61-63. We suggest initiating selection of crossmatched platelets by aggressively testing the alloimmunization status after multiple platelet exposures instead of obtaining bad CCI values after platelet transfusion for two times.

  • The sentence from "More alert (line 111)...to blood banks (line 112)", is improper and should be omitted.

Response: Thanks for your comment. The sentence on Line 111-112 was deleted.

  • The way to calculate the quantification of the reaction grade by ratioing and the confidence limits of readings in negative cases, previously recommended, are still lacking.

Response: Thanks for your comment. We emphasized our fELISA a qualitative test 4 times on Line 106, 297, 313, and 336. We already discussed its limitation to correlate the density ratios of fELISA with the SPRCA results in the second paragraph of discussion, Line 293-304. Regarding to the confidence limits of readings in negative cases, we already presented “…the average reactivity ratio of SPRCA-negative sera was 0.97 ± 0.17 (Figure 2).” on Line 232-233.

  • The validation of ZZAPped PLT preparation shelf life and the method to detect the integrity of the stored PLT samples, as previously recommended, are still missing.

Response: Thanks for your comment. The following paragraph was added in the discussion Line 306-316.“Briefly, a single batch of ZZAP-treated platelets was applied to all experiments being performed in two months, from 2021/01/11 to 2021/03/10. Both mock and negative control (NC) serum were included in each run of 10-well fELISA. After storing ZZAP-treated platelets at 4℃ for 2 months, the same 24 test samples were tested in three separate runs, two runs on 20210309 and one run on 20210310. Density value of mock control remain lower than NC serum (Mock/NC < 1). Final positive interpretation was defined as Sample/NC density ratio greater than 1. Twenty-two of 24 samples presented the same qualitative results with the SPRCA results. Only two samples were discrepant from the SPRCA results (bald and underlined). Therefore, we concluded the performance of fELISA remained consistent even using the platelets treated two months ago.

  • The sentence from "The pathophysiological (line 288)...to future study (line 290)" must be omitted.

Response: Thanks for your comment. The sentence on Line 290-292 was deleted.

  • In this v2 the described assay has been more clearly set as a screening method using a pool of 8 platelet concentrates. The details of the necessary organisation to collect and renew with time PLT concentrates from HLA-typed donors should be added in the discussion.

Response: Thanks for your comment. We added one sentence at the last of discussion, Line 342-344. “And the clinical applicability of our fELISA highly depends on obtaining platelets from 8 representative donors from blood establishments every two months because of the shelf-life limitation.

9) The entire sentence from "Our fELISA (line 328)...to ZZAP treatment (line 333)" should be omitted. The use of an assay designed to detect allo-immunisataion is not recommended nor validated in the autoimmune setting. This statement is improper, also because HIT and VITT are studied by validated specific assays (i.e. PF4 and Functional flow assays) and alloantibodies are absent in transfusion-naive ITP patients.

Response: Thanks for your comment. The sentence on Line 328-333 was deleted and change to the following “Our fELISA test is neither intended for detecting anti-platelet antibodies of transfusion-naïve ITP, HIT, and even VITT patients.” On Line 340-342.

Round 3

Reviewer 3 Report

I have appreciated the Authors' efforts and the care used to amend the last pending issues. 

Just a short comment: The paragraph concerning the validation of ZZAPped platelets shelf life, was included as suggested, but it has been placed in the DISCUSSION section. Please move the paragraph from line 305 to 316 in a specific subchapter of the RESULTS section (i.e. as section 3.1 due to its basic importance).

Author Response

Review Report (Reviewer 3)

I have appreciated the Authors' efforts and the care used to amend the last pending issues. 

Just a short comment: The paragraph concerning the validation of ZZAPped platelets shelf life, was included as suggested, but it has been placed in the DISCUSSION section. Please move the paragraph from line 305 to 316 in a specific subchapter of the RESULTS section (i.e. as section 3.1 due to its basic importance).

Response to reviewer: Appreciate your time and precious comment for our manuscript. As your suggestion, we moved the paragraph concerning the validation of ZZAPped platelets shelf life was to the RESULTS section 3.3 on Line 242~251. The section described detail results of 2 times fELISA for the same 24 samples using a single batch ZZAPed platelets preserved for two months.

And we will request and pay for the MDPI English editing service checking grammar, spelling, punctuation and some improvement of style after completing response to all comments of reviewers.